# Evolution of Nutritional Status after Early Nutritional Management in COVID-19 Hospitalized Patients

**DOI:** 10.3390/nu13072276

**Published:** 2021-06-30

**Authors:** Dorothée Bedock, Julie Couffignal, Pierre Bel Lassen, Leila Soares, Alexis Mathian, Jehane P. Fadlallah, Zahir Amoura, Jean-Michel Oppert, Pauline Faucher

**Affiliations:** 1Center for Research on Human Nutrition Ile-de-France (CRNH IdF), Nutrition Department, Institute of Cardiometabolism and Nutrition (ICAN), Assistance Publique-Hôpitaux de Paris (AP-HP), Pitié-Salpêtrière University Hospital, Sorbonne Université, 47-83 Boulevard de l’Hôpital, 75013 Paris, France; dorothee.bedock@aphp.fr (D.B.); julie.couffignal@aphp.fr (J.C.); pierre.bellassen@aphp.fr (P.B.L.); leila.soares@aphp.fr (L.S.); pauline.faucher@aphp.fr (P.F.); 2NutriOmics Team, INSERM UMRS U1166, Sorbonne Université, 75006 Paris, France; 3French National Referral Center for Systemic Lupus Erythematosus, Antiphospholipid Antibody Syndrome and Other Autoimmune Disorders, Centre d’Immunologie et des Maladies Infectieuses (CIMI-Paris), Medicine Department, Inserm UMRS, Assistance Publique–Hôpitaux de Paris, Pitié-Salpêtrière University Hospital, Sorbonne Université, 47-83 Boulevard de l’Hôpital, 75013 Paris, France; alexis.mathian@aphp.fr (A.M.); jehane.fadlallah@aphp.fr (J.P.F.); zahir.amoura@aphp.fr (Z.A.)

**Keywords:** COVID-19, malnutrition, pneumonia, SARS-Cov2, albumin, nutritional management

## Abstract

**Background & Aims**: SARS-CoV2 infection is associated with an increased risk of malnutrition. Although there are numerous screening and nutritional management protocols for malnutrition, only few studies have reported nutritional evolution after COVID-19. The objectives of this study were to describe the evolution of nutritional parameters between admission and 30 days after hospital discharge, and to determine predictive factors of poor nutritional outcome after recovery in adult COVID-19 patients. **Methods**: In this observational longitudinal study, we report findings after discharge in 91 out of 114 patients initially admitted for COVID-19 who received early nutritional management. Nutritional status was defined using GLIM criteria and compared between admission and day 30 after discharge. Baseline predictors of nutritional status at day 30 were assessed using logistic regression. **Results**: Thirty days after discharge, 28.6% of patients hospitalized for COVID-19 were malnourished, compared to 42.3% at admission. Half of malnourished patients (53%) at admission recovered a normal nutritional status after discharge. Weight trajectories were heterogeneous and differed if patients had been transferred to an intensive care unit (ICU) during hospitalization (*p* = 0.025). High oxygen requirement during hospitalization (invasive ventilation *p* = 0.016 (OR 8.3 [1.6–61.2]) and/or oxygen therapy over 5 L/min *p* = 0.021 (OR 3.2 [1.2–8.9]) were strong predictors of malnutrition one month after discharge. **Conclusions**: With early nutritional management, most patients hospitalized for COVID-19 improved nutritional parameters after discharge. These findings emphasize the importance of nutritional care in COVID-19 patients hospitalized in medicine departments, especially in those transferred from ICU.

## 1. Introduction

The recently described SARS-Cov2 infection (COVID-19) is responsible of severe pneumonia and high mortality rate. Several risk factors for severe COVID-19 have been identified including older age, male gender, metabolic comorbidities, but also malnutrition [1,2]. Correlation between nutritional status and severity or mortality of viral pneumonia had been described since the times of the 1918 influenza pandemic and more recently in other viral pneumonia including H1N1 virus [3,4]. Patients affected by COVID-19 have been shown to be at high risk of malnutrition [5,6,7,8]. Indeed, using composite nutritional scores such as the Nutrition Risk Screening tool (NRS-2002) or the Controlling Nutritional Status (CONUT) score, the risk of COVID-19 related malnutrition has been estimated to be between 77 and 88% [6,8,9]. Moreover, in a few recent studies using strictly defined anthropometric parameters based on GLIM criteria to define nutritional status [10], the prevalence of malnutrition in patients hospitalized for SARS-CoV2 was as high as 40 to 50% [8,11].

Origins of malnutrition in COVID-19 patients is likely because of reduced food intake, increased digestive and cutaneous losses (diarrhea, vomiting, sweating), olfactory and gustatory dysfunctions, hyper catabolism due to high level of inflammation and muscle wasting [10]. It is known that in hospitalized patients for medical or surgical disease, whatever they are, malnutrition is associated with short-term complications: greater hospital mortality, more frequent readmissions and longer hospital stays [12,13]. Unfavorable nutritional status may also affect long-term functional outcomes as described in survivors of acute respiratory distress syndrome (loss of muscle mass and gain of fat mass) [14]. Currently, little is known about consequences of malnutrition and the effects of nutritional care in COVID-19 patients.

Scientific societies in the field of nutrition have proposed protocols for early screening and management of malnutrition in patients hospitalized for COVID-19 [15,16,17,18]. However, only one study [19] has described short-term changes in nutritional status after nutritional management in such patients. To investigate further the evolution of malnutrition in COVID-19 patients, we implemented in April 2020 in our university hospital setting a nutritional protocol for care of patients seen during the first wave of the epidemics.

The objectives of the present study were to assess the effects of early nutritional management in hospitalized adult patients with COVID-19, comparing the prevalence and severity of malnutrition before and 30 days after hospital discharge. We then investigated the evolution of nutritional parameters one month after hospital discharge of these patients. Finally, we searched predictive factors of poor nutritional outcome after COVID-19 recovery.

## 2. Materials & Methods

### 2.1. Study Population

This observational longitudinal study included all adult COVID-19 patients admitted to the E3M Institute at Pitié-Salpêtrière hospital (Assistance-Publique-Hôpitaux de Paris, Paris, France) from March 21st 2020 to April 24th 2020. Patients were admitted from the emergency department, medical units or intensive care units [6]. Confirmed SARS-Cov2 infection was defined by a positive result of real-time reverse-transcriptase-polymerase-chain-reaction (RT-PCR) assay from nasal and pharyngeal swab specimens and/or evocative thoracic CT-scan damage. The protocol was approved by the Ethical Research Committee of Sorbonne University (Paris, France) (CER-2020-71). All data were collected in the context of care. Therefore, in accordance with French law (including the GPRD), an informed consent from the patient was not sought. Patients were informed that data from their medical records might be used for research in accordance with privacy rules and that they could express their refusal. No patient opposed.

A total of 114 COVID-19 patients were hospitalized and had complete nutritional data during that time period. Among these, 18 were admitted from an intensive care unit (ICU) whereas 96 were admitted as a first line of care in our standard medical unit. Six (5.3%) died during their stay and 17 (14.9%) were still hospitalized at follow-up. The present study reports the follow-up findings in 91 out of 114 patients (i.e., 78.8%) who were examined 30 days after hospital discharge. Mean length of stay was 9.7 days [1; 34].

### 2.2. Nutritional Assessment

During hospitalization, data on last known weight (i.e., weight within the past 6 months before admission for COVID-19, referred to in the text as “usual weight”) and height were self-reported by patients. Actual weight was measured on calibrated scales during their hospital stay allowing to calculate the percentage of weight loss from usual weight defined as ((usual weight–measured weight) * 100)/usual weight. BMI was calculated as measured weight (kg) divided by height (meter) squared. Albumin and transthyretin levels were measured at admission by routine lab procedures.

Diagnosis of malnutrition according to the GLIM criteria requires at least one etiologic criterion among reduced food intake or assimilation, inflammation or disease burden, and at least one phenotypic criterion among non-volitional weight loss, low BMI and reduced muscle mass [10]. In our setting, we considered that an etiologic criterion of malnutrition was always present due to COVID-19-induced inflammation. Additionally, in most cases, reduced food intake or assimilation, the other etiologic criterion, was present due to COVID-19 specific symptoms (anosmia, dysgueusia, nausea, vomiting, diarrhea). Phenotypic criteria were assessed using BMI and % weight loss from usual weight. We also used the GLIM criteria to categorize the severity of malnutrition [10]. Hospital registered dietitians performed individualized prescription when the patient did not eat the entire meal tray presented for lunch or dinner or when he/she had lost weight.

Based on international recommendations [7,8,9,10], we followed a standardized, individualized and pragmatic approach for nutritional management [15,16,17,18]. We differentiated 3 situations: (i) No malnutrition when the patient consumed his/her entire tray and had not lost weight; (ii) moderate malnutrition if the patient consumed between half to ¾ of the tray and/or had lost 5 to 15% of his/her weight ; (iii) severe malnutrition if the patient consumed less than half of his/her tray and/or had lost more than 15% of his/her weight. All meals were systematically enriched with calories and proteins (to reach an average daily intake of 2160 kcal instead of the usual 2000 kcal). Systematic prevention of refeeding syndrome was performed with either oral nutritional supplements or artificial nutrition in patients presenting moderate or severe malnutrition [15,20].

Thirty days after hospital discharge, 91 patients came back for a follow-up examination. On that day, a detailed medical re-assessment was conducted by both an internist and a specialist in clinical nutrition including medical history since discharge, current medical treatment, course of COVID-19 hospitalization, COVID-19 complications and residual symptoms. In addition, blood sampling was performed for routine clinical biological assessment including albumin and transthyretin levels, and follow-up weight (digital scale) and standing height were measured. Percent weight variation from usual and from hospitalization weight were calculated respectively as ((follow-up weight * 100)/usual weight) and ((follow-up weight * 100)/hospitalization weight). Body composition was assessed by bio-impedance analysis (Tanita TBF 300 GS; Tanita, France) [21]. Muscle function was assessed by handgrip testing (Electronic Hand Dynamometer EH101; Camry, Hong Kong). As performed during hospitalization of patients, nutritional status and malnutrition severity at follow-up were defined using GLIM criteria [10]. When needed after the medical examination, a dietary information sheet was provided to patients and further advice by a registered dietician was offered.

### 2.3. Assessment of Severity of COVID-19

Demographic features and comorbidities such as obesity, high blood pressure, diabetes, active smoking, chronic obstructive pulmonary disease, chronic kidney disease, chronic heart disease, known to be associated with COVID-19 severity [5,22] were screened during hospitalization. Following the same line, leucocytes, lymphocytes, polynuclear neutrophils, C-Reactive Protein (CRP), D-dimers levels were measured at admission [5,22]. Oxygen intake (L/min) at admission and during hospitalization, chest CT-scan pulmonary infiltrate, use of invasive or non-invasive ventilation, transfer to an intensive care unit (ICU) were also recorded to assess COVID-19 severity.

### 2.4. Statistical Analyses

Continuous variables were expressed as mean ± SD. Categorical variables were expressed as absolute values and percentages. Continuous variables with a non-parametric distribution were log-transformed before analysis. Weight change from usual weight (before COVID-19) at the time of admission and 30 days after hospital discharge was analysed in a mixed linear model in relation with the initial parameters of severity of COVID-19. Linear regression with analysis of variance (ANOVA) for continuous variables and Pearson’s chi-square (χ2) test or Fisher’s exact test for discrete variables were used to compare the different characteristics according to nutritional status. Relationships between baseline parameters and nutritional status 30 days after hospital discharge were investigated using a univariate logistic regression model. Measures of association between baseline variables and malnutrition at day 30 after discharge were performed by computing odds ratios [23]. Statistical tests were considered significant if *p* < 0.05. All statistical analyses were conducted using R studio software version 1.2.1335 (http://www.r-project.org, (accessed on 8 April 2019)).

## 3. Results

Table 1 presents the general characteristics and nutritional parameters of patients according to their GLIM nutritional status at day 30 after discharge. For the whole sample at admission, subjects were on average middle-aged but with a large age range (from 20 to 96 y). Subjects displayed several comorbidities: one-third with diabetes, one-fourth with chronic heart disease and one-fifth with chronic kidney disease. One-fourth of the sample was obese (BMI ≥ 30 kg/m^2^). Thirty days after hospital discharge, 26 out of 91 patients (28.6%) of patients were malnourished (and only 6 (6.6%) severely malnourished). There was no significant difference in muscle mass and handgrip strength between well-nourished and malnourished patients at day 30 after discharge (Table 1).

There was high inter individual variability in weight change over time (Figure 1).

At day 30 after discharge, weight change from admission in the whole sample was on average +2.7% compared to admission. Weight change from admission differed significantly according to nutritional status at day 30 after discharge (−2.0%, +1.4% and +3.6% in severely, moderately malnourished and well-nourished patients, respectively). Weight trajectories differed according to whether patients had stayed in ICU during COVID-19 hospital management. On average, 30 days after hospital discharge, patients did not return to their usual weight. This was more pronounced in patients who had stayed in ICU during their hospitalization (Figure 2).

Changes in nutritional status according to GLIM criteria at admission and at day 30 after discharge are shown on Figure 3. At admission, 42.3% of the patients were malnourished (moderate or severe malnutrition combined). At day 30 after discharge, 53% of initially malnourished patients had normal nutritional status. Among those who had a normal nutritional status at admission, 14.3% were moderately malnourished at day 30 after discharge. Among patients who were moderately malnourished at admission, 2 patients became severely malnourished at day 30 after discharge (Figure 3).

Patients malnourished compared to those with normal nutritional status at day 30 after hospital discharge had higher weight loss at admission, had received more frequently invasive ventilation and over 5 L/min O_2_ replacement therapy and had higher initial polynuclear neutrophils level. There was no significant difference in muscle mass and handgrip strength between well-nourished and malnourished patients at day 30 after discharge (Table 1). Use of invasive ventilation and receiving O_2_ over 5 L/min were strong predictors of malnutrition 30 days after discharge (Table 2).

## 4. Discussion

In a series of 91 consecutive COVID-19 adult patients with various comorbidities, we were able to collect a range of phenotypical (BMI, weight loss) and biological (albumin, CRP) data at the beginning of hospitalization for COVID-19 and one month after hospital discharge. Our analysis reveals that 28.6% of patients hospitalized for COVID-19 were malnourished 30 days after hospital discharge, compared to 42.3% at admission. In parallel with early nutritional management, more than half of malnourished patients at admission recovered a normal nutritional status at day 30 after discharge. Weight trajectories were heterogeneous and differed according to whether patients had been transferred to ICU during hospital management. High oxygen requirement during hospitalization (invasive ventilation and/or oxygen therapy over 5 L/min) for COVID-19 were strong predictors of malnutrition one month after admission.

A main finding of this study is that almost half of patients with malnutrition at admission recovered a normal nutritional status 30 days after discharge. All patients received optimized nutritional management as adapted from international and French guidelines on nutritional screening and support [16,17,18], including systematic prevention of refeeding syndrome. These data demonstrate that recovering from malnutrition after COVID-19 is possible with early nutritional support. Nutritional evaluation after COVID-19 is as yet not well defined. In a recent scoping review, Mechanick et al. did not find any research regarding nutrition risk assessment during outpatient follow-up after COVID-19 [24]. Only one study (not included in that scoping review) presented nutritional data at follow-up of patients hospitalized for COVID-19: in 185 patients admitted for COVID-19 in an emergency department, De Lorenzo et al. reported that at a median follow-up of 23 days, 5.4% of patients were malnourished [19]. In our study, 28.6% of patients were malnourished 30 days after hospital discharge. We can formulate several explanations. First, De Lorenzo et al. used body weight change and Mini Nutritional Assessment (MNA) screening tool to assess malnutrition whereas we used strict GLIM criteria [10]. Moreover, our patients exhibited more comorbidities (such as diabetes, obesity, chronic organ failure), which are known independent risk factors of malnutrition [2,5,23,25,26,27,28,29]. Indeed, in our sample, 24.2% of patients suffered from chronic heart disease vs. 6.5% in the De Lorenzo study, 17.6% had chronic kidney disease vs. 2.4%, and 7.7% had chronic obstructive pulmonary disease vs. 1.6%. Furthermore, ICU transfer during a COVID-19 event is associated higher risk of malnutrition [11] and in our population, this was the case for 33% of patients compared to 3.2% of hospitalized patients in the report of De Lorenzo et al. Another potential factor which could contribute to malnutrition at day 30 after discharge is the persistence of olfactory and gustatory dysfunctions. Indeed, smell and/or taste disturbances could lead to reduced food intake after general symptoms of COVID-19 resolution. Lechien et al. estimated that 56% of patients have persistent olfactory dysfunction over the days following the resolution of the COVID-19 general clinical manifestations [29].

An important finding of this study is that we were able to describe individual nutritional parameters (weight loss, weight trajectories, biological parameters) after the acute phase of COVID-19 infection. At admission, we observed in our patients a median weight loss of −5.4% comparing to their (self-reported) usual weight. At day 30 after hospital discharge, even if individual weight trajectories were heterogeneous, patients had regained weight on average although they were still at −1.8% of their usual weight. This is similar to the results shown by De Lorenzo et al. for hospitalized patients (−2% of the usual weight at follow-up) [19]. There was also a significant difference in weight trajectories according to whether patients had stayed in ICU during COVID-19 hospital management. Invasive ventilation is a strong predictor of malnutrition at follow-up and is only used in ICU. Another predictor of malnutrition was the use of O2 over 5 L/min, a sign of respiratory severity of COVID-19. Indeed, acute respiratory complications required ICU management and stay, which is associated with high risk of malnutrition because of hypercatabolism and food intake impairment, and with severe loss of skeletal muscle mass and function [16]. At last, in our study, all 91 patients had normal level of albumin one month after hospital discharge and two thirds improved (i.e., went from severe to moderate malnutrition) or normalized their nutritional status, which supports the efficiency of an early nutritional support in hospitalized COVID-19 patients. However, some patients had worsened their nutritional status, which can have several explanations: some had obesity and perhaps did not wish to regain their initial weight, some had psychological and/or social fragilities that occurred after their hospitalization, which could participate to their poor nutritional status at follow-up.

An important strength of our study in COVID-19 patients is that we were able to obtain follow-up nutritional data 30 days after hospitalization. We also used the specific metrics defined by GLIM [10] for malnutrition diagnosis at admission and 30 days after hospitalization even though these parameters are difficult to collect due to the lack of scales and in consideration of the hygienic precautions required in this particular setting [15]. Some limitations need to be mentioned. Our sample size possibly led to underestimate the prevalence of malnutrition 30 days after COVID-19 and thus, led to insufficient power to determine factors associated with malnutrition at day 30. We were not able to rigorously test the efficiency of our early nutritional management in the absence of a control group, which would have not been ethical in these conditions. Moreover, one of the phenotypic criteria defining malnutrition in GLIM is the loss of muscle mass and function. Even though we found no significant difference in muscle mass and handgrip strength between well-nourished and malnourished patients at day 30, it would have been of interest to have such data at admission. It is likely that, as previously observed in the context of other respiratory diseases, sarcopenia may heavily influence the course and outcomes of COVID-19 infection [30]. To be able to assess the effects of nutritional management on sarcopenia would be important in such setting [31,32].

## 5. Conclusions

In conclusion, we observed that recovering from malnutrition after COVID-19 hospitalization is possible with early nutritional management. In these patients hospitalized for COVID-19, weight trajectories differed when a transfer to ICU was needed during hospital management. Use of invasive ventilation and/or use of oxygen therapy over 5 L/min during hospitalization for COVID-19 were strong predictors of malnutrition one month after discharge. These findings point to the importance of nutritional management during hospitalization and after hospital discharge for patients with severe respiratory COVID-19, especially in those transferred from ICU. Further studies are needed to more fully assess the long-term impact of nutritional management during COVID-19 hospitalization, including sarcopenia.

## Figures and Tables

**Figure 1 nutrients-13-02276-f001:**
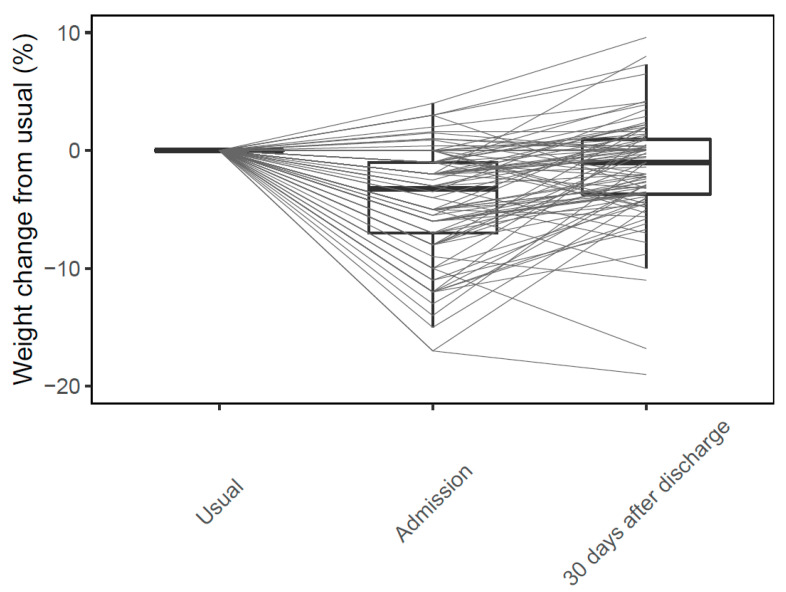
Individual weight change (%) in COVID-19 patients between usual weight, hospital admission and 30 days after discharge.

**Figure 2 nutrients-13-02276-f002:**
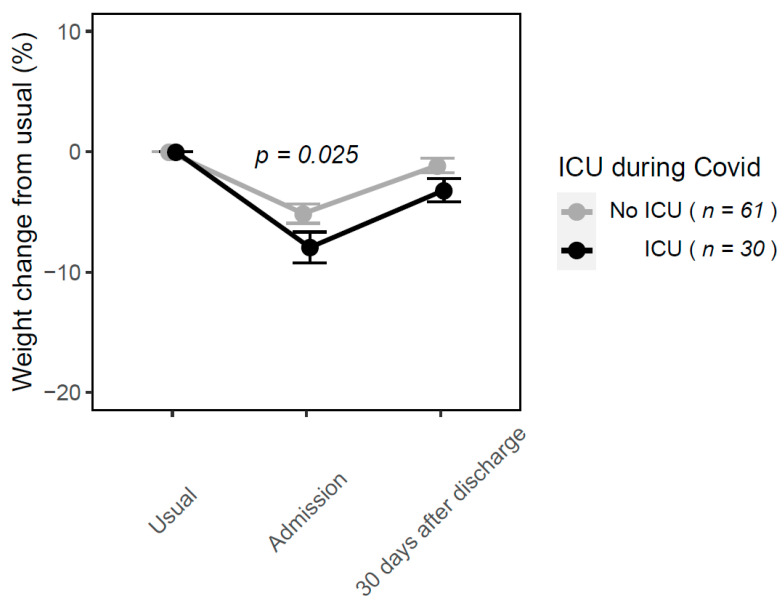
Mean weight change (%) in COVID-19 patients depending on whether patients were transferred to ICU or not during hospital stay.

**Figure 3 nutrients-13-02276-f003:**
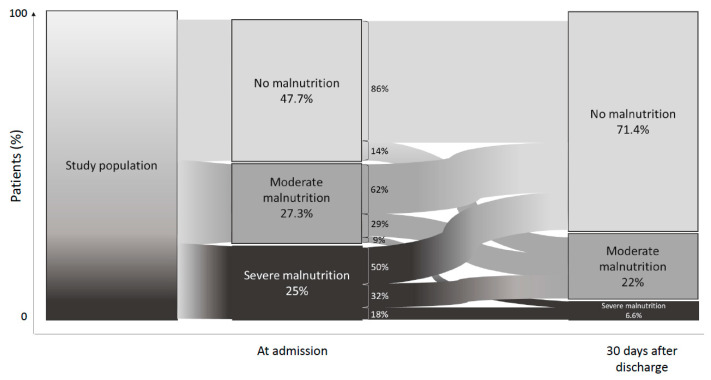
Evolution of nutritional status according to GLIM criteria between admission and 30 days after hospital discharge in COVID-19 patients.

**Table 1 nutrients-13-02276-t001:** General characteristics and nutritional parameters of COVID-19 patients according to GLIM nutritional status 30 days after hospital discharge.

	[All]	0	1	2	*p*
	n = 91	n = 65	n = 20	n = 6	Value
**At admission**					
Age (years)	57.7 (15.6)	57.5 (14.4)	54.4 (18.3)	70.2 (14.9)	0.09
Male, *n* (%)	55 (60.4)	40 (61.5)	12 (60.0)	3 (50.0)	0.87
High blood pressure, *n* (%)	46 (50.5)	5 (53.8)	8 (40.0)	3 (50.0)	0.55
Diabetes, *n* (%)	30 (33.0)	24 (36.9)	6 (30.0)	0 (0.0)	0.22
Active smoking, *n* (%)	4 (4.8)	1 (1.6)	3 (18.8)	0 (0.0)	0.05
Chronic obstructive pulmonary disease, *n* (%)	7 (7.7)	4 (6.2)	1 (5.0)	2 (33.3)	0.08
Chronic kidney disease, *n* (%)	16 (17.6)	11 (16.9)	4 (20.0)	1 (16.7)	0.89
Chronic heart disease, *n* (%)	22 (24.2)	13 (20.0)	6 (30.0)	3 (50.0)	0.15
Weight at admission (kg)	76.3 (17.3)	77.8 (14.5)	75.0 (21.5)	64.3 (25.8)	0.18
Usual weight (kg)	80.4 (18.0)	80.9 (15.0)	81.3 (21.8)	71.0 (32.4)	0.42
Weight change from usual (%)	−5.4 (5.7)	−4.3 (5.3)	−8.0 (5.9)	−7.0 (7.6)	0.03
BMI (kg/m^2^)	26.5 (5.1)	26.7 (4.5)	26.5 (6.0)	24.1 (7.1)	0.50
Obesity, *n* (% BMI ≥ 30 kg/m^2^)	21 (23.9)	14 (22.6)	6 (30.0)	1 (16.7)	0.83
Albumin (g/L)	30.3 (5.6)	30.0 (5.7)	30.8 (6.0)	32.2 (1.7)	0.62
Transthyretin (g/L)	0.1 (0.1)	0.1 (0.1)	0.2 (0.1)	0.2 (0.1)	0.05
**At 30 days after discharge**					
Weight (kg)	78.8 (17.4)	81.1 (15.2)	76.4 (19.5)	62.4 (25.1)	0.03
Weight change from usual (%)	−1.8 (4.9)	0.2 (3.7)	−5.7 (2.9)	−10.2 (6.3)	<0.01
Weight change from admission (%)	2.7 (4.3)	3.6 (3.8)	1.4 (5.0)	−2.0 (3.8)	<0.01
BMI (kg/m^2^)	27.5 (4.9)	28.0 (4.5)	27.0 (5.4)	23.3 (6.5)	0.07
Obesity, *n* (% BMI ≥ 30 kg/m^2^)	64 (71.1)	46 (71.9)	13 (65.0)	5 (83.3)	0.73
Weight change from usual (kg)	−1.6 (4.4)	0.1 (3.0)	−4.9 (3.0)	−8.6 (7.7)	<0.01
Muscle mass (kg)	55.0 (12.3)	55.9 (12.3)	53.2 (11.1)	49.4 (18.5)	0.45
Handgrip strength (kg)	33.4 (19.6)	35.2 (20.9)	28.3 (10.1)	28.1 (30.7)	0.35
Albumin (g/L)	42.1 (4.7)	42.2 (5.1)	41.4 (3.6)	43.0 (4.7)	0.75
Transthyretin (g/L)	0.3 (0.1)	0.3 (0.1)	0.3 (0.2)	0.3 (0.1)	0.64
**COVID parameters**					
Invasive ventilation, *n* (%)	7 (7.9)	2 (3.1)	4 (21.1)	1 (20.0)	0.02
Non-invasive ventilation, *n* (%)	20 (22.2)	11 (16.9)	7 (36.8)	2 (33.3)	0.11
O2 over 5 L/min during COVID, *n* (%)	7 (7.7)	1 (1.5)	5 (25.0)	1 (16.7)	<0.01
Intensive Care Unit during COVID, *n* (%)	30 (33.0)	18 (27.7)	10 (50.0)	2 (33.3)	0.16
CT pulmonary infiltrate over 50%, *n* (%)	9 (11.5)	7 (12.5)	1 (5.9)	1 (20.0)	0.45
Leucocytes (G/L)	6.9 (3.9)	6.2 (2.9)	9.3 (6.1)	7.2 (1.8)	0.01
Lymphocytes (G/L)	1.1 (0.5)	1.0 (0.5)	1.1 (0.6)	1.4 (0.2)	0.25
Polynuclear Neutrophils (G/L)	5.3 (3.8)	4.6 (2.6)	8.1 (6.7)	5.2 (1.9)	0.01
CRP (mg/L)	91.1 (73.3)	100.3 (77.5)	63.5 (50.2)	61.2 (56.4)	0.16
D dimers (µg/L)	1410.4 (2035.1)	1607.2 (2291.4)	795.0 (818.7)	986.7 (371.0)	0.45

Results are expressed as mean (SD) for continuous data and *n* (%) for categorical data. *p* values shown result from analysis of variance (ANOVA) for continuous data and Chi2 or Fisher’s exact test for categorical data between the 3 categories of nutritional status. BMI: Body Mass Index.

**Table 2 nutrients-13-02276-t002:** Baseline predictors of malnutrition (moderate or severe) according to GLIM nutritional status 30 days after hospital discharge.

Baseline Predictors of Malnutrition at Day 30	OR [CI 95%]	*p* Value
**At admission**		
Age	1.0 [0.9–1.0]	0.894
Male	0.9 [0.3–2.2]	0.735
High blood pressure	0.6 [0.2–1.6]	0.322
Diabetes	0.5 [0.2–1.4]	0.209
Active smoking	9.5 [1.1–197.8]	0.058
Chronic obstructive pulmonary disease	2.0 [0.4–9.7]	0.391
Chronic kidney disease	1.2 [0.3–3.6]	0.794
Chronic heart disease	2.1 [0.8–5.8]	0.146
Weight at admission	1.0 [0.9–1.0]	0.197
Usual weight	1.0 [0.9–1.0]	0.634
Weight change from usual	1.1 [1.0–1.2]	0.013
BMI	1.0 [0.9–1.1]	0.524
Obesity (BMI ≥ 30 kg/m^2^)	1.3 [0.4–3.6]	0.663
Albumin	1.0 [1.0–1.1]	0.408
Transthyretin	1.1 [1.0–1.2]	0.061
**COVID parameters**		
Invasive ventilation	8.3 [1.6–61.2]	0.016
Non-invasive ventilation	2.8 [1.0–7.9]	0.056
O2 over 5 L/min during COVID	3.2 [1.2–8.9]	0.021
Intensive Care Unit during COVID	2.2 [0.9–5.8]	0.094
CT pulmonary infiltrate over 50%	0.7 [1.0–3.2]	0.673
Leucocytes	1.2 [1.0–1.4]	0.017
Lymphocytes	1.9 [0.7–5.3]	0.190
Polynuclear Neutrophils	1.2 [1.0–1.4]	0.024
CRP	1.0 [0.9–1.0]	0.060
D dimers	1.0 [0.9–1.0]	0.181

## Data Availability

Data available on request due to restrictions eg privacy or ethical. The data presented in this study are available on request from the corresponding author. The data are not publicly available due to the format of consent.

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
