# Peer review of "Evolution of Nutritional Status after Early Nutritional Management in COVID-19 Hospitalized Patients"

_nutrients, 2021, doi:10.3390/nu13072276_

Round 1

Reviewer 1 Report

Dear authors

I have some comments to your study:

1) the introduction is brief and does not properly justify the importance and interest of the study.

Please develop the introduction appropriately to provide an adequate theoretical framework for the study. 

2) line 64-65. Authors write "Patients were admitted from the 64 emergency department, medical units or intensive care units, as previously described (6).", but where is described in the manuscript. Please include this information to clarify your procedure to future readers.

3) Reading your manuscript it was impossible for to me find any mention of the ethical issues about the development of your study. You have used data obtained from human and doesn't matter that these data come from the routinary diagnostic test. Patients, ever, must be informed about the study and the objectives that pursuit the researchers. 
Please, include a section where you detailed all aspects associated with the Ethical concerns of your study, including the Clinical Research Ethical Committee approval (with its codification). 

It is a mandatory point to accept your manuscript.

4) Statistical analysis

 4.1.- First of all I would like to ask why they have decided not to show the non-significant data. It is important in science to show the negative as well as the "positive". I believe, that showing such data would give a clearer picture of the study and its conclusions, maybe including or modifying tables to include all data analyzed.

(i. e.: (lines 277-278 )"we found no significant difference in muscle mass and handgrip strength 277 between well-nourished and malnourished patients at day 30"  , but I have no access to the data that shows what you declare in your affirmation. )

4.2.- Furthermore, I have noted that the results include the Odds Ratio but you have not indicated that measures of concordance/association, between variables, will be performed in the statistical analyses. Please include this information.

And please, justify (from a methodological or mathematical point of view) why you use the OR instead of the RR in a prospective and longitudinal study?

Reviewer 2 Report

The topic dealt with in this manuscript is of absolute interest because it emphasizes an aspect that is often not so considered as that of hospital malnutrition.This condition can affect the prognostic evolution of the underlying disease and determine health consequences in the short to medium term.

The authors describe the evolution of nutritional parameters between admission and 30 days after hospital discharge, and to determine predictive factors of poor nutritional outcome after recovery in adult COVID-19 patients. This is an observational longitudinal study that concerns a very current aspect, namely postdischarge malnutrition in adult patients hospitalized for SARS -coV2 infection. The criteria used, GLIM criteria, represent a valid method to define a framework of malnutrition. The authors demonstrated that a high oxygen requirement during hospitalization (invasive ventilation p=0.016 (OR 8.3 [1.6-61.2]) and/or oxygen therapy over 5L/min p=0.021 (OR 3.2 [1.2-8.9]) were strong predictors of malnutrition one month after discharge.    

The results are well presented, interesting and well discussed so I believe that the manuscript will be published in this form.

Round 2

Reviewer 1 Report

.